# Optimal Lottery Tickets via SUBSETSUM: Logarithmic Over-Parameterization is Sufficient

**Ankit Pensia**[*]
University of Wisconsin-Madison
ankitp@cs.wisc.edu

**Shashank Rajput**[*]
University of Wisconsin-Madison
rajput3@wisc.edu

**Alliot Nagle**
University of Wisconsin-Madison
acnagle@wisc.edu

**Harit Vishwakarma**
University of Wisconsin-Madison
hvishwakarma@cs.wisc.edu

**Dimitris Papailiopoulos**
University of Wisconsin-Madison
dimitris@papail.io

## Abstract

The strong *lottery ticket hypothesis* (LTH) postulates that one can approximate any target neural network by only pruning the weights of a sufficiently over-parameterized random network. A recent work by Malach et al. [1] establishes the first theoretical analysis for the strong LTH: one can provably approximate a neural network of width $d$ and depth $l$, by pruning a random one that is a factor $O(d^4 l^2)$ wider and twice as deep. This polynomial over-parameterization requirement is at odds with recent experimental research that achieves good approximation with networks that are a small factor wider than the target. In this work, we close the gap and offer an exponential improvement to the over-parameterization requirement for the existence of lottery tickets. We show that any target network of width $d$ and depth $l$ can be approximated by pruning a random network that is a factor $O(\log(dl))$ wider and twice as deep. Our analysis heavily relies on connecting pruning random ReLU networks to random instances of the SUBSETSUM problem. We then show that this logarithmic over-parameterization is essentially optimal for constant depth networks. Finally, we verify several of our theoretical insights with experiments.

## 1 Introduction

Many of the recent unprecedented successes of machine learning can be partially attributed to state-of-the-art neural network architectures that come with up to tens of billions of trainable parameters. Although test accuracy is one of the gold standards in choosing one of these architectures, in many applications having a "compressed" model is of practical interest, due to typically reduced energy, memory, and computational footprint [2–15]. Such a compressed form can be achieved by either modifying the architecture to be leaner in terms of the number of weights, or by starting with a high-accuracy network and pruning it down to one that is sparse in some representation domain, while not sacrificing much of the original network's accuracy. A rich and long body of research work shows that one can prune a large network to a tiny fraction of its size, while maintaining (or sometimes even improving) its original accuracy [2–15].

---

[*]Authors contributed equally to this paper and are listed alphabetically.

Although network pruning dates back at least to the 80s [16–19], there has been a recent flurry of results that introduce sophisticated pruning, sparsification, and quantization techniques which lead to significantly compressed model representations that attain state-of-the-art accuracy [2, 14, 15]. Several of these pruning methods require many rounds of pruning and retraining, resulting in a time-consuming and hard to tune iterative meta-algorithm.

Could we avoid this pruning and retraining cycle, by "winning" the weight initialization lottery, with a ticket that puts stochastic gradient descent (SGD) on a path to, not only accurate, but also very sparse neural networks? Frankle and Carbin [20] define *lottery tickets* as sparse subnetworks of a randomly initialized network, that if trained to full accuracy just once, can reach the performance of the fully-trained but dense target model. If these lottery tickets can be found efficiently, then the computational burden of pruning and retraining can be avoided.

Several works build and expand on the *lottery ticket hypothesis* (LTH) that was introduced by Frankle and Carbin [20]. Zhou et al. [21] experimentally analyze different strategies for pruning. Frankle et al. [22] relate the existence of lottery tickets to a notion of stability of networks for SGD. Cosentino et al. [23] show that lottery tickets are amenable to adversarial training, which can lead to both sparse and robust neural networks. Soelen et al. [24] show that the winning tickets from one task are transferable to another related task. Sabatelli et al. [25] show further that the trained winning tickets can be transferred with minimal retraining on new tasks, and sometimes may even generalize better than models trained from scratch specifically for the new task.

Along this literature, a striking finding was reported by Ramanujan et al. [26] and Wang et al. [27]: one does not even need to train the lottery tickets to get high test accuracy; they find that high-accuracy, sparse models simply reside within larger random networks, and appropriate pruning can reveal them. However, these high-accuracy lottery tickets do not exist in plain sight, and finding them is in itself a challenging (as a matter of fact NP-Hard) computational task. Still, the mere existence of these random substructures is interesting, and one may wonder whether it is a universal phenomenon.

The phenomenon corroborated by the findings of Ramanujan et al. [26] is referred to as the *strong lottery ticket hypothesis*. Recently, Malach et al. [1] proved the strong LTH for fully connected networks with ReLU activations. In particular, they show that one can approximate any target neural network, by pruning a sufficiently over-parameterized network of random weights. The degree of over-parameterization, i.e., how much larger this random network has to be, was bounded by a polynomial term with regards to the width $d$ and depth $l$ of the target network. Specifically, their analysis requires the random network to be of width $\widetilde{O}(d^5 l^2/\epsilon^2)$ and depth $2l$, to allow for a pruning that leads to an $\epsilon$-approximation with regards to the output of the target network, for any input in a bounded domain. They also show that the required width can be improved to $\widetilde{O}(d^2 l^2/\epsilon^2)$ under some sparsity assumptions on the input. Although polynomial, the required degree of over-parameterization is still too demanding, and it is unclear that it explains the experimental results that corroborate the strong LTH. For example, it does not reflect the findings in Ramanujan et al. [26] that only seem to require a constant factor over-parameterization, e.g., a randomly initialized Wide ResNet50, can be pruned to a model that has the accuracy of a fully trained ResNet34. In this work, our goal is to address the following question:

> *What is the required over-parameterization so that a network of random*
> *weights can be pruned to approximate a smaller target network?*

Towards this goal, we identify the crucial step in the proof strategy of Malach et al. [1] that leads to the polynomial factor requirement on the per-layer over-parameterization. Say that one wants to approximate a weight $w^* \in [-1, 1]$ with a random number drawn from a distribution, e.g., Uniform($[-1, 1]$). If we draw $n = O(1/\epsilon)$ i.i.d. samples,

$$X_1, \ldots, X_n \sim \text{Uniform}([-1, 1]),$$

then one of these $X_i$'s will be $\epsilon$ close to $w^*$, with constant probability. In a way, this random sampling generates an $\epsilon$-net, i.e., a set of numbers such that any weight in $[-1, 1]$ is $\epsilon$ close to one of these $n$ samples. Pruning that set down to a single number, i.e., selecting the point closest to $w^*$, leads to an $\epsilon$ approximation for a single weight. At the cost of a polynomial overhead on the number of samples $n$, one can appropriately apply this to every weight of a given layer, and then all layers of the network, in order to obtain a uniform approximation result for every possible input of the neural network.

Perhaps the surprising fact is that one can achieve the same approximation, with exponentially smaller number of samples, while still using a pruning algorithm. The idea is that instead of approximating the target weight $w^*$ with only one of the $n$ samples $X_1, \ldots, X_n$, one could add a subset of them to get a better approximation. Indeed, when $n = O(\log(1/\epsilon))$, then, with high probability, there exists a subset $S \subseteq [1, \ldots, n]$, such that

$$\left| w^* - \sum_{i \in S} X_i \right| \leq \epsilon.$$

Hence, approximating $w^*$ by the best subset sum $\sum_{i \in S} X_i$, offers an exponential improvement on the sample complexity requirement compared to approximating it with one of the $X_i$'s.

The above approximation result for random $X_i$'s is drawn from a line of work on the random SUBSET-SUM problem [28–31]. In particular, Lueker [31] showed that one can achieve an $\epsilon$-approximation of any target number $t \in [-1, 1]$, with only $O(\log(1/\epsilon))$ i.i.d. samples from any distribution that contains a uniform distribution on $[-1, 1]$, by using the solution to the following (NP-hard in general) problem

$$\min_S \left| t - \sum_{i \in S} X_i \right|.$$

Adapting this result to ReLU activation functions is precisely what allows us to get an exponential improvement on the over-parameterization required for the strong LTH to be true.

**Our Contributions:** In this work, by adapting the random SUBSETSUM results of Lueker [31] to ReLU activation functions, we offer an exponential improvement on the over-parameterization required for the strong LTH to be true. In particular we establish the following result.

**Theorem 1.** *(informal) A randomly initialized network with width $O(d \log(dl/\min\{\epsilon, \delta\}))$ and depth $2l$, with probability at least $1 - \delta$, can be pruned to approximate any neural network with width $d$ and depth $l$, up to error $\epsilon$.*

The formal statement of Theorem 1 is provided in Section 3. Note that in the above result, no training is required to obtain the sparser network within the random model, i.e., *pruning is all you need*. Further, note that we guarantee a good approximation to *any* network of fixed width and depth by pruning a *single* larger network that is logarithmically wider. That is, the set of networks obtained by pruning the larger random network amounts to an $\epsilon$-net with regards to the smaller target networks.

We then show that this logarithmic over-parameterization is essentially optimal for constant depth networks. Specifically, we provide a lower bound for 2-layered networks that matches the upper bound proposed in Theorem 1, up to logarithmic terms with regards to the width (see Theorem 2 in Section 4 for a formal statement):

**Theorem 2.** *(informal) There exists a 2-layer neural network with width $d$ which cannot be approximated to error within $\epsilon$ by pruning a randomly initialized 2-layer network, unless the random network has width at least $\Omega(d \log(1/\epsilon))$.*

To the best of our knowledge, the only prior work that proves the validity of the strong LTH is Malach et al. [1]. However, as discussed earlier, their result gives an upper bound of $\widetilde{O}(d^5 l^2)$ on the required width of the random networked to be pruned, which we improve to $O(d \log(dl))$. A concurrent and independent work by Orseau et al. [32] also prove a version of the strong LTH, where they use a hyperbolic distribution for initialization, and require the over-parameterized network to have width $O(d^2 \log(dl))$. Although Orseau et al. [32] prove their result for the hyperbolic distribution, they conjecture that it also holds for the uniform distribution. Theorem 1 proves this conjecture with an improved guarantee that a $O(d \log(dl))$ wide network suffices (which is smaller by a factor of $d$). Furthermore, our result holds for a broad class of distributions beyond the uniform distribution (see Remark 1).

**Notation** We use lower-case letters to represent scalars, e.g., we may use $w$ to denote the weight of a single link between two neurons. We use bold lower-case letters to denote vectors, for example, $\mathbf{v}, \mathbf{u}, \mathbf{v}_1, \mathbf{v}_2$. The $i$-th coordinate of the vector $\mathbf{v}$ is denoted as $v_i$. Finally, matrices are denoted by bold upper-case letters. For a vector $\mathbf{v}$, we use $\|\mathbf{v}\|$ to denote its $\ell_2$ norm. If a matrix $\mathbf{W}$ has dimension $d_1 \times d_2$, we say $\mathbf{W} \in \mathbb{R}^{d_1 \times d_2}$. The operator norm of a $d_1 \times d_2$ dimensional matrix $\mathbf{M}$ is defined as $\|\mathbf{M}\| = \sup_{\mathbf{x} \in \mathbb{R}^{d_2}, \|\mathbf{x}\|=1} \|\mathbf{M}\mathbf{x}\|$. We denote the uniform distribution on $[a, b]$ by $U[a, b]$. We use $c, C$ to denote positive absolute constants, which may vary from place to place, and their exact values can be inferred from the proof details

## 2 Preliminaries and Setup

In this work, our goal is to approximate a target network $f(\mathbf{x})$ by pruning a larger network $g(\mathbf{x})$, where $\mathbf{x} \in \mathbb{R}^{d_0}$. The target network $f$ is a fully-connected, ReLU neural network of the following form

$$f(\mathbf{x}) = \mathbf{W}_l \sigma(\mathbf{W}_{l-1} \dots \sigma(\mathbf{W}_1 \mathbf{x})), \tag{1}$$

where $\mathbf{W}_i$ has dimension $d_i \times d_{i-1}$, $\mathbf{x} \in \mathbb{R}^{d_0}$, and $\sigma(\cdot)$ is the ReLU activation that is, $\sigma(x) = x \cdot \mathbf{1}_{x \geq 0}$. A second, larger network $g(\mathbf{x})$ is of the following form

$$g(\mathbf{x}) = \mathbf{M}_{2l} \sigma(\mathbf{M}_{2l-1} \dots \sigma(\mathbf{M}_1 \mathbf{x}). \tag{2}$$

Our goal is to obtain a pruned version of $g$ by eliminating weights, i.e.,

$$\hat{g}(\mathbf{x}) = (\mathbf{S}_{2l} \odot \mathbf{M}_{2l}) \sigma((\mathbf{S}_{2l-1} \odot \mathbf{M}_{2l-1}) \dots \sigma((\mathbf{S}_1 \odot \mathbf{M}_1)\mathbf{x})), \tag{3}$$

where each $\mathbf{S}_i$ is a binary (pruning) matrix, with the same dimension as $\mathbf{M}_i$, and $\odot$ represents element-wise product between matrices. Our objective is to obtain a good approximation while controlling the size of $g(\cdot)$, i.e., the width of $\mathbf{M}_i$'s. In this work we only prune neuron weights, and not entire neurons. We refer the reader to Malach et al. [1] for the differences between the two approaches: *pruning weights* and *pruning neurons*.

We consider the case where the weight matrices $\mathbf{M}_{2l}, \dots, \mathbf{M}_1$ of $g(\cdot)$ are randomly initialized. In particular, each element of the matrices is an independent sample from $U[-1, 1]$.

The error metric that we use is the uniform approximation over the normed-ball, i.e., $\hat{g}$ is $\epsilon$-close to $f$ in the following sense:

$$\max_{\mathbf{x} \in \mathbb{R}^{d_0}: \|\mathbf{x}\| \leq 1} \|f(\mathbf{x}) - \hat{g}(\mathbf{x})\| \leq \epsilon.$$

Observe that a result of this kind can be generalized from the domain $\{\|\mathbf{x}\| \leq 1\}$ to an arbitrarily large radius $r$, $\{\|\mathbf{x}\| \leq r\}$, by scaling $\epsilon$ appropriately. It is necessary, though, to consider only bounded domains because ReLU neural networks are positive-homogeneous ($f(\alpha \mathbf{x}) = \alpha f(\mathbf{x})$ for $\alpha \geq 0$) and thus, any non-zero error can be made arbitrarily large for unbounded domains.

## 3 Lottery Tickets via Subset Sum

We now present our results for approximating a target network by pruning a sufficiently over-parameterized neural network. In fact, we prove that a single random, logarithmically over-parameterized neural network can be pruned to approximate any neural network of a fixed architecture, with high probability. We define $\mathcal{F}$ to be the set of target ReLU neural networks $f$ such that (i) $f : \mathbf{R}^{d_0} \to \mathbf{R}^{d_l}$, (ii) $f$ has depth $l$ (iii) weight matrix of $i$-th layer has dimension $d_i \times d_{i-1}$ and spectral norm at most 1. That is,

$$\mathcal{F} = \{f : f(\mathbf{x}) = \mathbf{W}_l \sigma(\mathbf{W}_{l-1} \dots \sigma(\mathbf{W}_1 \mathbf{x})), \forall i \ \mathbf{W}_i \in \mathbb{R}^{d_i \times d_{i-1}} \text{ and } \|\mathbf{W}_i\| \leq 1\} \tag{4}$$

We prove that a randomly initialized neural network

$$g(\mathbf{x}) = \mathbf{M}_{2l} \sigma(\mathbf{M}_{2l-1} \dots \sigma(\mathbf{M}_1 \mathbf{x})),$$

which has $2l$ layers and layer widths $\log \frac{d_{i-1} d_i l}{\min\{\epsilon, \delta\}}$ times the corresponding layer widths of $\mathcal{F}$, with probability $1 - \delta$, can approximate any neural network in $\mathcal{F}$ up to error $\epsilon$. For simplicity, we state our results for matrices with spectral norm at most 1, but they can readily be generalized to arbitrary norm bounds.

**Theorem 1.** *Let $\mathcal{F}$ be as defined in Eq. (4). Consider a randomly initialized $2l$-layered neural network*

$$g(\mathbf{x}) = \mathbf{M}_{2l} \sigma(\mathbf{M}_{2l-1} \dots \sigma(\mathbf{M}_1 \mathbf{x})),$$

*where every weight is drawn from $U[-1, 1]$, $\mathbf{M}_{2i}$ has dimension*

$$d_i \times C d_{i-1} \log \frac{d_{i-1} d_i l}{\min\{\epsilon, \delta\}},$$

*and $\mathbf{M}_{2i-1}$ has dimension*

$$C d_{i-1} \log \frac{d_{i-1} d_i l}{\min\{\epsilon, \delta\}} \times d_{i-1}.$$

*Then, with probability at least $1 - \delta$, for every $f \in \mathcal{F}$,*

$$\min_{\mathbf{S}_i \in \{0,1\}^{d_i \times d_{i-1}}} \sup_{\|\mathbf{x}\| \leq 1} \|f(\mathbf{x}) - (\mathbf{S}_{2l} \odot \mathbf{M}_{2l})\sigma((\mathbf{S}_{2l-1} \odot \mathbf{M}_{2l-1})\ldots\sigma((\mathbf{S}_1 \odot \mathbf{M}_1)\mathbf{x}))\| < \epsilon.$$

Note that the above result offers a uniform approximation guarantee for all networks in $\mathcal{F}$ by only pruning a single over-parameterized network $g$. In this sense if $\mathcal{G}$ is the set of all pruned versions of the base neural network $g(\mathbf{x})$, then our guarantee states that, with probability $1 - \delta$,

$$\sup_{f \in \mathcal{F}} \min_{\hat{g}(\mathbf{x}) \in \mathcal{G}} \sup_{\|\mathbf{x}\| \leq 1} \|f(\mathbf{x}) - \hat{g}(\mathbf{x})\| < \epsilon.$$

We note two generalizations of Theorem 1 in the remarks below:

**Remark 1.** *Although Theorem 1 is stated for initialization with the uniform distribution, Leuker's result allows us to extend it to a wide family of distributions. Suppose $P$ is a univariate distribution that contains a uniform distribution in the following sense: there exists a distribution $Q$, a constant $\alpha \in (0,1]$, and a constant $c > 0$ such that $P = (1 - \alpha)Q + \alpha U[-c, c]$. Then a random network initialized with $P$ also satisfies the guarantee of Theorem 1, up to constants depending on $\alpha$ and $c$. For example, the standard normal distribution and the Laplace distribution with zero mean and unit variance satisfy this condition.*

**Remark 2.** *Theorem 1 assumes that both the target network and the over-parameterized network have ReLU activations. This assumption can be relaxed to some extent. Looking at Eq.* (13) *(in the proof of Theorem 1, Appendix A.3), we see that the target network $f$ can have any activation function, as long as it is 1-Lipschitz. The over-parameterized network $g$ would still have $2l$ layers, where every odd layer would have a ReLU (or linear) activation and every even layer would have the same activation as the target network.*

In Subsection 3.1, we illustrate the connection between the SUBSETSUM problem and approximating a single weight via pruning by considering a linear network. Later, in Subsection 3.2, we show how to do the same by pruning a ReLU network. Towards the end of Subsection 3.2, we outline a proof sketch of Theorem 1. The complete proof of Theorem 1 can be found in Appendix A.

## 3.1 Single Link: Pruning a Linear Network

We now explain our approximation scheme for a single link weight by pruning a random two-layered *linear* neural network. Let the scalar target function be $f(x) = w \cdot x$, where $|w| \leq 0.5$. To make the task simpler, let us assume (just for this subsection) that the second layer is deterministic, and has weights that are all equal to 1. Thus, the over-parameterized neural network $g(\cdot)$, that we will prune, has the following linear architecture:

$$g(x) = \mathbf{1}^T \mathbf{a} x = \sum_{i=1}^{n} a_i x,$$

where $\mathbf{1}$ is all-ones vector, $\mathbf{a} = [a_1, \ldots, a_n]^T$, and the weights $a_i$ are sampled from $U[-1, 1]$. Figure 1b shows a visual representation of this network.

Then, the question is how large does $n$ (the width of the random network) need to be so that we can approximate $wx$ by pruning weights in $\mathbf{a}$. As $|x| \leq 1$, we see that it is equivalent to following:

$$\mathbb{P}\left(\exists S \subseteq [n] : \left|w - \sum_{i \in S} a_i\right| \leq \epsilon\right) \geq 1 - \delta, \tag{5}$$

where the probability is taken over the randomness in $a_i$'s. Note that this condition is tightly related to a random instance of the subset sum problem, i.e., $\min_{S \subseteq \{1,\ldots,n\}} \left|w - \sum_{i \in S} a_i\right|$. This problem was studied by Lueker [31], who obtains the following result:

**Theorem 3.** *(Corollary 2.5 [31]) Let $X_1, \ldots, X_n$ be i.i.d. uniform over $[-1, 1]$, where $n \geq C \log \frac{2}{\min\{\epsilon, \delta\}}$. Then, with probability at least $1 - \delta$, we have*

$$\forall z \in [-0.5, 0.5], \qquad \exists S \subset [n] \text{ such that } |z - \sum_{i \in S} X_i| \leq \epsilon.$$

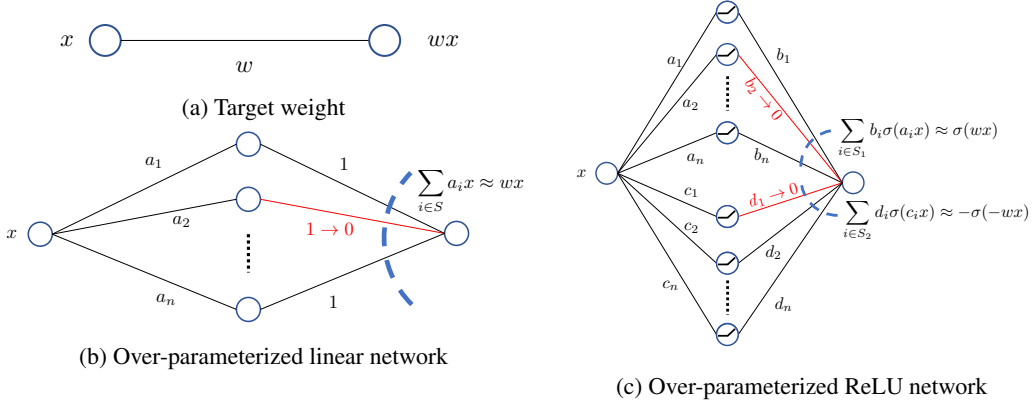

Figure 1: (a) Target network is a linear univariate $wx$. (b) A simplified construction, where we approximate the weight $w$ by pruning an over-parameterized two-layered linear network with deterministic second layer of all 1s. We only prune the second layer by setting weights to 0. (c) Our construction from the proof of Lemma 1 that uses ReLU units. Our construction adds an additional hidden layer of size $2n = O(\log(1/\epsilon))$. Without loss of generality, assume $w \geq 0$. As the hidden layer also has ReLU activation, we make use of the identity $wx = \sigma(wx) - \sigma(-wx)$ and approximate the two terms separately. We prune the network such that the (i) top half of the network $(a_i, b_i)$ approximates $\sigma(wx)$, (ii) and the bottom half $(c_i, d_i)$ approximates $-\sigma(-wx)$. We only prune the weights in the second layer (shown in red) for a technical reason which helps us reuse the weights in the first layer subsequently in the proof of Theorem 1.

Lueker in [31], establishes this theorem by a beautiful and intricate proof that employs concentration of martingales, and is crucial for the proof of our upper bound. Using Theorem 3, we see that if $n \geq C \log(1/\min\{\epsilon, \delta\})$, then Eq. (5) holds. We can prune the network by simply setting $a_i$ to 0 for $i \notin S$. Equivalently, we could instead prune the output layer at indices that are not in $S$, as shown in Figure 1b. Note that the dependence on $\epsilon$ is only logarithmic, which leads to only logarithmic dependence on width in Theorem 1.

## 3.2 Single Link: Pruning a ReLU Network

We now show how the ideas from Subsection 3.1 can be extended to random ReLU networks.

**Lemma 1.** *(Approximating a weight) Let $g : \mathbb{R} \to \mathbb{R}$ be a randomly initialized network of the form $g(x) = \mathbf{v}^T \sigma(\mathbf{u}x)$, where $\mathbf{v}, \mathbf{u} \in \mathbb{R}^{2n}$, $n \geq C \log \frac{2}{\epsilon}$, and all $v_i$, $u_i$'s are drawn i.i.d. from $U[-1, 1]$. Then with probability at least $1 - \epsilon$,*

$$\forall w \in [-1, 1], \quad \exists \ \mathbf{s} \in \{0, 1\}^{2n} : \sup_{x:|x|\leq 1} |wx - (\mathbf{v} \odot \mathbf{s})^T \sigma(\mathbf{u}x)| < \epsilon.$$

*Proof.* We decompose $wx$ as $wx = \sigma(wx) - \sigma(-wx)$ and WLOG assume $w \geq 0$. Let

$$\mathbf{v} = \begin{bmatrix} \mathbf{b} \\ \mathbf{d} \end{bmatrix}, \mathbf{u} = \begin{bmatrix} \mathbf{a} \\ \mathbf{c} \end{bmatrix}, \mathbf{s} = \begin{bmatrix} \mathbf{s}_1 \\ \mathbf{s}_2 \end{bmatrix},$$

where $\mathbf{a}, \mathbf{b}, \mathbf{c}, \mathbf{d} \in \mathbb{R}^n, \mathbf{s}_1, \mathbf{s}_2 \in \{0, 1\}^n$. Thus, $(\mathbf{v} \odot \mathbf{s})^T \sigma(\mathbf{u}x) = (\mathbf{b} \odot \mathbf{s}_1)^T \sigma(\mathbf{a}x) + (\mathbf{d} \odot \mathbf{s}_2)^T \sigma(\mathbf{c}x)$. See Figure 1c for a diagram.

**Step 0: Equivalence between pruning u and v.** Note that $(\mathbf{v} \odot \mathbf{s})^T \sigma(\mathbf{u}x) = \mathbf{v}^T \sigma((\mathbf{s} \odot \mathbf{u})x)$. Thus, in the the following construction, we will prune $\mathbf{u}$ ($\mathbf{a}$ and $\mathbf{c}$) instead of $\mathbf{v}$, for simplicity.

**Step 1: Pre-processing a.** Let $\mathbf{a}^+ = \max\{\mathbf{0}, \mathbf{a}\}$ be the vector obtained by pruning all the negative entries of $\mathbf{a}$. Thus, $\mathbf{a}^+$ contains $n$ i.i.d. random variables from the mixture distribution: $(1/2)P_0 + (1/2)U[0, 1]$, where $P_0$ is the degenerate distribution at 0.

Since $w \geq 0$, then for $x \leq 0$ we have that $\sigma(wx) = \mathbf{b}^T \sigma(\mathbf{a}^+x) = 0$. Moreover, further pruning of $\mathbf{a}^+$ would not affect this equality for $x \leq 0$. We thus focus our attention on $x > 0$ in steps 1 and 2. Therefore, we get that $\sigma(wx) = wx$ and $\mathbf{b}^T \sigma(\mathbf{a}^+x) = \mathbf{b}^T \mathbf{a}^+x = \sum_i b_i a_i^+ x$.

**Step 2: Pruning $a$ via SUBSETSUM.** Consider the random variable $Z_i = b_i a_i^+$. We show that Theorem 3 also holds for $Z_i$'s (See Corollary 1 and Corollary 2 in Appendix C). Therefore, as long as

$n \geq C \log 2/\epsilon$, with probability $1 - \epsilon/2$, we can choose a subset of $\{Z_1, Z_2, \ldots, Z_n\}$ to approximate $w$ up to $\epsilon$. That is with probability $1 - \epsilon/2$,

$$\forall w \in [0,1], \quad \exists \mathbf{s}_1 \in \{0,1\}^n : \quad |w - \mathbf{b}^T(\mathbf{s}_1 \odot \mathbf{a}^+)| < \epsilon/2,$$

and because $|x| \leq 1$, we can uniformly bound the error between the linear function $wx$ and the pruned version of the network. Therefore, with probability $1 - \epsilon/2$,

$$\forall w \in [0,1], \quad \exists \mathbf{s}_1 \in \{0,1\}^n : \quad \sup_{x \in [-1,1]} |\sigma(wx) - \mathbf{b}^T\sigma((\mathbf{s}_1 \odot \mathbf{a}^+)x)| < \epsilon/2, \qquad (6)$$

where we use that for $x < 0$, $\sigma(wx) = \mathbf{b}^T\sigma((\mathbf{s}_1 \cdot \mathbf{a}^+))x = 0$.

**Step 3: Pre-processing c.** We now turn our attention to $x \leq 0$ with a similar procedure as steps 1 and 2. We begin by pre-processing $\mathbf{c}$. Let $\mathbf{c}^- = \min\{\mathbf{0}, \mathbf{c}\}$ be the vector obtained by pruning the positive entries of $\mathbf{c}$. Therefore, $\mathbf{c}^-$ contains $n$ i.i.d. random variables from the mixture distribution: $(1/2)P_0 + (1/2)U[-1,0]$, where $P_0$ is the degenerate distribution at 0.

**Step 4: Pruning c via SUBSETSUM.** For $x \geq 0$, we have that $\sigma(-wx) = 0$ and $\mathbf{d}^T\sigma(\mathbf{c}^-x) = 0$. Moreover, pruning $\mathbf{c}^-$ further does not affect the equality. Thus, we only consider the case $x < 0$.

For $x < 0$, we get that $-\sigma(-wx) = -(-wx) = wx$ and also $\sigma(\mathbf{c}^-x) = \mathbf{c}^-x$. Thus $\mathbf{d}^T\sigma(\mathbf{c}^-x) = (\mathbf{d}^T\mathbf{c}^-)x$. Observe that the $b_i a_i^+$ and $d_i c_i^-$ have the exact same distribution and thus similar to the step 2 above, as long as $n \geq C \log 2/\epsilon$, with probability at least $1 - \epsilon/2$,

$$\forall w \in [0,1], \quad \exists \mathbf{s}_2 \in \{0,1\}^n : \quad \sup_{x \in [-1,1]} |-\sigma(-wx) - \mathbf{d}^T\sigma((\mathbf{s}_2 \odot \mathbf{c}^-)x)| < \epsilon/2, \qquad (7)$$

where we use that $wx = -\sigma(-wx)$ and $\mathbf{d}^T(\mathbf{s}_2 \odot \mathbf{c}^-)x = \mathbf{d}^T\sigma((\mathbf{s}_2 \odot \mathbf{c}^-)x)$ for $x \leq 0$.

**Step 5: Tying it all together.** Recall that $w \geq 0$. By a union bound, we assume that Equations (6) and (7) hold with probability at least $1 - \epsilon$. On that event, we get that

$$\inf_{\mathbf{s} \in \{0,1\}^{2n}} \sup_{|x| \leq 1} |wx - \mathbf{v}^T\sigma((\mathbf{u} \odot \mathbf{s})x)| = \inf_{\mathbf{s}_1, \mathbf{s}_2} \sup_{|x| \leq 1} |wx - \mathbf{b}^T\sigma((\mathbf{s}_1 \odot \mathbf{a})x) - \mathbf{d}^T\sigma((\mathbf{s}_2 \odot \mathbf{c})x)|$$

$$\leq \inf_{\mathbf{s}_1, \mathbf{s}_2} \sup_{|x| \leq 1} |wx - \mathbf{b}^T\sigma((\mathbf{s}_1 \odot \mathbf{a}^+)x) - \mathbf{d}^T\sigma((\mathbf{s}_2 \odot \mathbf{c}^-)x)| \qquad \text{(Using steps 1 and 3)}$$

$$= \inf_{\mathbf{s}_1, \mathbf{s}_2} \sup_{|x| \leq 1} |\sigma(wx) - \sigma(-wx) - \mathbf{b}^T\sigma((\mathbf{s}_1 \odot \mathbf{a}^+)x) - \mathbf{d}^T\sigma((\mathbf{s}_2 \odot \mathbf{c}^-)x)|$$

$$\leq \inf_{\mathbf{s}_1} \sup_{|x| \leq 1} |\sigma(wx) - \mathbf{b}^T\sigma((\mathbf{s}_1 \odot \mathbf{a}^+)x)| + \inf_{\mathbf{s}_2} \sup_{|x| \leq 1} |-\sigma(-wx) - \mathbf{d}^T\sigma((\mathbf{s}_2 \odot \mathbf{c}^-)x)|$$

$$\leq \epsilon,$$

where the last step uses (6) and (7). $\qquad \square$

**Proof Sketch of Theorem 1** Lemma 1 states that we can approximate a single link with a $O(\log(1/\epsilon))$ wide network. Then we show that we can approximate a neuron with $d$ weights using $O(d \log(d/\epsilon))$ wide network. Moreover, reusing weights allows us to approximate a whole layer of $d$ neurons with a $O(d \log(d/\epsilon))$ wide network. Finally, we show that if we can approximate each layer in the target network individually, we also get a good approximation as a whole.

## 4 Lower Bound by Parameter Counting

We now state the lower bound for the required over-parameterization by showing that even approximating a linear network requires blow up of width by $\log(1/\epsilon)$. For a matrix $\mathbf{W}$, we can express the linear function $\mathbf{Wx}$ as a ReLU network $h_{\mathbf{W}}$ (see Eq. (8)). Let $\mathcal{F}$ be the set of neural networks, that represent linear functions with spectral norm at most 1, i.e.,

$$\mathcal{F} := \{h_{\mathbf{W}} : \mathbf{W} \in \mathbf{R}^{d \times d} : \|\mathbf{W}\| \leq 1\}, \quad \text{where} \quad h_{\mathbf{W}}(\mathbf{x}) = [\mathbf{I} \quad -\mathbf{I}]\,\sigma\left(\begin{bmatrix} \mathbf{W} \\ -\mathbf{W} \end{bmatrix}\mathbf{x}\right). \qquad (8)$$

We prove that if a random network (with arbitrary distribution) approximates every $h_W \in \mathcal{F}$ with probability at least $0.5$, then the random network needs at least $d^2 \log(1/\epsilon)$ parameters. For a two-layered network, this means that the width must be at least $d \log(1/\epsilon)$. Note that our lower bound

does not require a uniform approximation over $\mathcal{F}$, which is achieved by Theorem 1 with $d \log(d/\epsilon)$ width. Therefore, Theorem 2 shows a lower bound to the version of the *lottery ticket hypothesis* considered by Malach et al. [1], whereas Theorem 1 shows an upper bound for a stronger version of the *LTH*. Formally, we have the following theorem:

**Theorem 2.** *Consider a neural network, $g : \mathbb{R}^d \rightarrow \mathbb{R}^d$ of the form $g(\mathbf{x}) = \mathbf{M}_l \sigma(\mathbf{M}_{l-1} \ldots \sigma(\mathbf{M}_1 \mathbf{x}))$, with arbitrary distributions on $\mathbf{M}_1, \ldots, \mathbf{M}_l$. Let $\mathcal{G}$ be the set of neural networks that can be formed by pruning $g$, i.e., $\mathcal{G} := \{\widehat{g} : \widehat{g}(\mathbf{x}) = (\mathbf{S}_l \odot \mathbf{M}_l)\sigma(\ldots \sigma((\mathbf{S}_1 \odot \mathbf{M}_1)\mathbf{x})), \text{ where } \mathbf{S}_1, \ldots \mathbf{S}_1 \text{ are pruning matrices}\}$. Let $\mathcal{F}$ be as defined in Eq. (8). If the following statement holds:*

$$\forall h \in \mathcal{F}, \mathbb{P}\left(\exists g' \in \mathcal{G} : \sup_{\mathbf{x}:\|\mathbf{x}\|\leq 1} \|h(\mathbf{x}) - g'(\mathbf{x})\| < \epsilon\right) \geq \frac{1}{2}, \tag{9}$$

*then $g(\mathbf{x})$ has $\Omega(d^2 \log(1/\epsilon))$ parameters. Further, if $l = 2$, then width of $g(\mathbf{x})$ is $\Omega(d \log(1/\epsilon))$.*

**Remark 3.** *Note that Theorem 2 focuses on approximating (linear) multivariate functions, which is an important building block in approximating multilayer neural networks. At the same time, our bounds are tight only for networks with constant depth. Generalizing Theorem 2 for deeper networks would require a better understanding of the increase in representation power of neural networks with depth. We leave further investigation of both of these questions for future work.*

**Proof Sketch:** Our proof strategy is a counting argument, which shows that $|\mathcal{G}|$ has to be large deterministically. Particularly, there exists a matrix $\mathbf{W}$ that is far from all the pruned networks in $\mathcal{G}$ with high probability, unless $|\mathcal{G}|$ is large enough. Together with the fact that $|\mathcal{G}|$ scales with the number of parameters, we get the desired lower bound. See Appendix B for the complete proof.

## 5 Experiments

We verify our results empirically by approximating a target network via SUBSETSUM in Experiment 1, and by pruning a sufficiently over-parameterized neural network that implements the structures in Figures 1b and 1c in Experiment 2. In both setups, we benchmark on the MNIST [33] dataset, and all training and pruning is accomplished with cosine annealing learning rate decay [34] on a batch size 64 with momentum 0.9 and weight decay 0.0005.

**Experiment 1: SUBSETSUM.** We approximate a two-layer, 500 hidden node target network with a final test set accuracy of 97.19%. Every weight was approximated in this network with a subset sum of $n = C \log_2(\frac{1}{\epsilon}) \approx 21$ coefficients, for $\epsilon = 0.01$ and $C = 3$. To solve SUBSETSUM more efficiently, we implement the following mixed integer program (MIP) for every weight $w$ in the target network and solve it using Gurobi's [35] MIP solver:

$$\min_{x_1,\ldots,x_n} \left|w - \sum_{i=1}^n a_i x_i\right| \quad \text{subject to} \quad \left|w - \sum_{i=1}^n a_i x_i\right| \leq \epsilon, \quad x_i \in \{0,1\} \,\forall i = 1,\ldots,n \,,$$
$$a_i \in [l, u] \,\forall i = 1,\ldots,n,$$

where $l$ and $u$ are the bounds of the uniform coefficient distribution and $\epsilon$ is sufficiently large. Every set of $a_i$ coefficients is unique to the approximation of $w$, and these coefficients are drawn uniformly from a range which is fine-tuned for the target network. We recommend that the set $[l, u]$ be large enough to at least contain all weights in the target network, making the solution to any SUBSETSUM problem less likely to be infeasible. When the bounds are close to the minimum and maximum weight values in the network, we find that we can decrease $n$, either by increasing $\epsilon$ or decreasing $C$, thereby reducing time complexity. Since the solver finds the optimal solution, most weights in our approximated network were well below 0.01 error, and our approximated network maintained the 97.19% test set accuracy. The $397,000$ weights in our target network were approximated with $3,725,871$ coefficients in 21.5 hours on 36 cores of a c5.18xlarge AWS EC2 instance. Such a running time is attributed to solving many instances of this nontrivial combinatorial problem.

**Experiment 2: Pruning Random Networks.** We train baseline networks, including two-layer and four-layer fully connected networks with 500 hidden nodes per layer (learning rate 0.1 for 10 epochs) and LeNet5 (learning rate 0.01 for 50 epochs). In Figure 4, we show the result of implementing the

structure in Figure 1c (with and without ReLU activation) in each of these networks, and compare the result with their respective baselines and wide-network counterparts. More specifically, we compare the results of pruning our structure with pruning a wider, random network such that the number of parameters is approximately equal to the number of parameters in the network with our structure. We use the `edge-popup` [26] algorithm to prune the networks, which finds a subnetwork in each of these two architectures without training the weights. Since our structure in Figure 1c is well-defined only for fully connected layers, we prune LeNet5 by randomly initializing and freezing the convolutional filters and pruning the fully connected layers. The weights and scores in the pruned networks are initialized with a Kaiming Normal [36] and Kaiming Uniform distribution, respectively. The LeNet5 networks are pruned with learning rate $0.01$; the fully connected networks utilize a learning rate of $0.1$. Experiments are run on these pruned architectures for $10, 20, 30, 50,$ and $100$ epochs, and the maximum accuracy for each architecture for a particular number of parameters is then selected. We vary the number of parameters in each network by adjusting the $[0, 1]$ sparsity parameter.

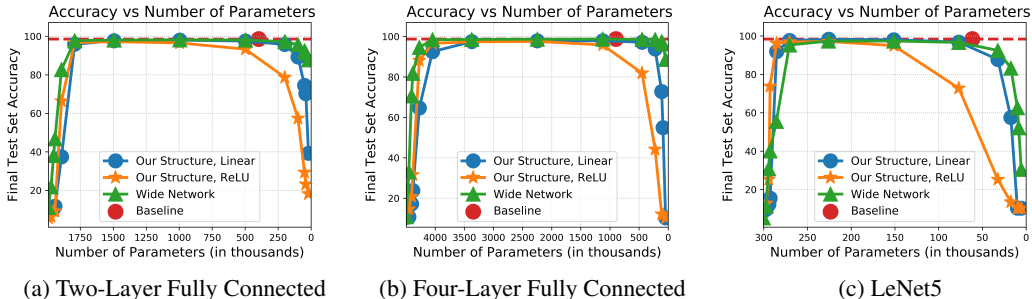

(a) Two-Layer Fully Connected     (b) Four-Layer Fully Connected     (c) LeNet5

Figure 2: Performance of pruning our structure shown in Figure 1c, with and without ReLU activation, using 5 $(a_i, b_i)$ coefficient pairs per weight. The number of parameters is a function of the sparsity of each network. Results are benchmarked on MNIST.

Note that the use of additional ReLU activations leads to worse performance than the use of an identity mapping when the number of parameters is small. We posit that the additional sparsification from ReLU degrades the performance of our already sparsified network.

The findings from this second experiment indicate that the performance of a pruning algorithm can vary with regards to its approximation capacity of the target network, depending on the network topology. Note that the different network topologies all contain, approximately, the same number of weights, hence should lead to similar approximations. However we see that the choice of architecture plays an important role in the performance and efficiency of a given pruning algorithm.

## 6 Discussion

In this paper we establish a tight version of the strong lottery ticket hypothesis: there always exist subnetworks of randomly initialized over-parameterized networks that can come close to the accuracy of a target network; further this can be achieved by random networks that are only a logarithmic factor wider than the original network. Our results are enabled by a very interesting paper on the random subset sum problem from Lueker [31], and the essential building block of our analysis is to show that a linear function $f(x) = \sum_{i=1}^{d} w_i x_i$ on $d$ parameters, can be approximated by selecting a subset of the coefficients of a random one that has $d \log(dl/\epsilon)$ parameters.

Our current work focuses on general fully connected networks. It would be interesting to extend the results to convolutional neural networks. Other interesting structures that come up in neural networks are sparsity and low-rank weight matrices. This leads to the question of whether we can leverage the additional structure in the target network to improve our results. An interesting question from a computational point of view is whether our analysis gives insights to improve the existing pruning algorithms [26]. As remarked in Malach et al. [1], the strong LTH implies that pruning an over-parameterized network to obtain good accuracy is NP-Hard in the worst case. It is an interesting future direction to find efficient algorithms for pruning which provably work under mild assumptions on the data.

## Broader Impact

As discussed in the Introduction, our results establish that we can "train" a neural network by only pruning a slightly larger network. As shown in Strubell et al. [37], training a single deep model (including hyper-parameter optimization and experimentation) has the carbon footprint equivalent to that of four cars through their lifetime, motivating the search for a more efficient training algorithm. Pruning is a radically different way of optimization as compared to the usual gradient based one, and recent works show that either using it alone or in tandem with conventional optimization techniques can lead to good performance [20, 26, 27]. Moreover, a sparse network is also useful when the models are deployed for inference. One of the major benefits of pruning is that sparser models can have smaller memory and computational requirements, leading in some cases to less energy consumption. As a result, pruned networks are useful in resource-constrained settings and have smaller carbon footprint. Nonetheless, the pruning algorithms, if proved to be successful, will need to go through the same scrutiny as the existing optimization algorithms such as robustness to adversarial examples [23].

Another contribution, which is more subtle, is that we connect the SUBSETSUM problem with pruning of neural networks and their optimization in general. We believe that both these fields can benefit from the existing literature of the SUBSETSUM problem [28–31]. We use the result by Lueker [31] which says that we only need a logarithmic sized set of random numbers on a bounded domain to approximate any number in that domain with high accuracy and high probability. It would be interesting to see the Machine Learning community apply these kind of results to other theoretical and practical problems.

## Acknowledgments and Disclosure of Funding

DP wants to thank Costis Daskalakis and Alex Dimakis for early discussions of the problem during NeurIPS2019 in Vancouver, BC.

This research is supported by an NSF CAREER Award #1844951, a Sony Faculty Innovation Award, an AFOSR & AFRL Center of Excellence Award FA9550-18-1-0166, and an NSF TRIPODS Award #1740707.

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
