[Supplementary Material]

# A  Proof of the upper bound

**Complete proof of the Theorem 1**  In the following subsections, we hierarchically build the construction for our proof of Theorem 1. We have shown how we approximate a single weight in Subsection 3.2. This first step is slightly different than the sketch above, in the sense that we approximate a single weight with a ReLU random network, rather than a linear one. We then approximate a single ReLU neuron in Subsection A.1, and a single layer in Subsection A.2. Finally, we approximate the whole network in Subsection A.3, which completes the proof of Theorem 1.

## A.1  Approximating a single neuron

In this subsection we prove the following lemma on approximating a (univariate) linear function $\mathbf{w}^T\mathbf{x}$, which highlights the main idea in approximating a (multivariate) linear function $\mathbf{Wx}$ (see Lemma 3 in Subsection A.2).

**Lemma 2.** *(Approximating a univariate linear function) Consider a randomly initialized neural network $g(\mathbf{x}) = \mathbf{v}^T\sigma(\mathbf{Mx})$ with $\mathbf{x} \in \mathbb{R}^d$ such that $\mathbf{M} \in \mathbb{R}^{Cd\log\frac{d}{\epsilon}\times d}$ and $\mathbf{v} \in \mathbb{R}^{Cd\log\frac{d}{\epsilon}}$, where each weight is initialized independently from the distribution $U[-1, 1]$.*

*Let $\widehat{g}(x) = (\mathbf{s}\odot\mathbf{v})^T\sigma((\mathbf{T}\odot\mathbf{M})\mathbf{x})$ be the pruned network for a choice of binary vector $\mathbf{s}$ and matrix $\mathbf{T}$. If $f_{\mathbf{w}}(\mathbf{x}) = \mathbf{w}^T\mathbf{x}$ be the linear function, then with probability at least $1 - \epsilon$,*

$$\forall\mathbf{w} : \|\mathbf{w}\|_\infty \leq 1, \exists \quad \mathbf{s}, \mathbf{T} : \quad \sup_{\mathbf{x}:\|\mathbf{x}\|_\infty\leq 1} \|f_{\mathbf{w}}(\mathbf{x}) - \widehat{g}(\mathbf{x})\| < \epsilon.$$

*Proof.* We will approximate $\mathbf{w}^T\mathbf{x}$ coordinate-wise. See Figure 3 for illustration.

Figure 3: Approximating a single neuron $\sigma(\mathbf{w}^T x)$: A diagram showing our construction to approximate a single neuron $\sigma(\mathbf{w}^T x)$. We construct the first hidden layer with $d$ blocks (shown in blue), where each block contains $k = O\left(\log\frac{d}{\epsilon}\right)$ neurons. We first pre-process the weights by pruning the first layer so that it has a block structure as shown. For ease of visualization, we only show two connections per block, i.e., each neuron in the $i^{\text{th}}$ block is connected to $x_i$ and (before pruning) the output neuron. We then use Lemma 1 to show that second layer can be pruned so that $i^{\text{th}}$ block approximates $w_i x_i$. Overall, the construction approximates $\mathbf{w}^T\mathbf{x}$. Note that, after an initial pre-processing of the first layer, we only prune the second layer so that we can re-use the weights to approximate other neurons in a layer.

**Step 1: Pre-processing** $\mathbf{M}$   We first begin by pruning $\mathbf{M}$ to create a block-diagonal matrix $\mathbf{M}'$. Specifically, we create $\mathbf{M}'$ by only keep the following non-zero entries:

$$\mathbf{M}' = \begin{bmatrix} \mathbf{u}_1 & 0 & \dots & 0 \\ 0 & \mathbf{u}_2 & \dots & 0 \\ \vdots & \vdots & \dots & 0 \\ 0 & 0 & \dots & \mathbf{u}_d \end{bmatrix}, \qquad \text{where } \mathbf{u}_i \in \mathbb{R}^{C \log\left(\frac{d}{\epsilon}\right)}$$

We choose the binary matrix $\mathbf{T}$ to be such that $\mathbf{M}' = \mathbf{T} \odot \mathbf{M}$. We also decompose $\mathbf{v}$ and $\mathbf{s}$ as

$$\mathbf{s} = \begin{bmatrix} \mathbf{s}_1 \\ \mathbf{s}_2 \\ \vdots \\ \mathbf{s}_d \end{bmatrix}, \qquad \mathbf{v} = \begin{bmatrix} \mathbf{v}_1 \\ \mathbf{v}_2 \\ \vdots \\ \mathbf{v}_d \end{bmatrix}, \text{ where } \mathbf{s}_i, \mathbf{v}_i \in \mathbb{R}^{C \log\left(\frac{d}{\epsilon}\right)}.$$

Using this notation, we can express our network as the following:

$$(\mathbf{s} \odot \mathbf{v})^T \sigma(\mathbf{M}'\mathbf{x}) = \sum_{i=1}^{d} (\mathbf{s}_i \odot \mathbf{v}_i)^T \sigma(\mathbf{u}_i x_i). \tag{10}$$

**Step 2: Pruning** $u$   Let $n = C \log(d/\epsilon)$ and define the event $E_{i,\epsilon}$ be the following event from the Lemma 1:

$$E_{i,\epsilon} := \left\{ \sup_{w \in [-1,1]} \inf_{\mathbf{s}_i \in \{0,1\}^n} \sup_{x:|x|\leq 1} |wx - (\mathbf{v}_i \odot \mathbf{s}_i)^T \sigma(\mathbf{u}_i x)| \leq \epsilon \right\}$$

Define the event $E_\epsilon := \bigcap_i E_{i,\epsilon}$, the intersection of all the events. We consider the event $E_{\frac{\epsilon}{d}}$, where the approximation parameter is $\frac{\epsilon}{d}$. For each $i$, Lemma 1 shows that event $E_{i,\frac{\epsilon}{d}}$ holds with probability at least $1 - \frac{\epsilon}{d}$ because the dimension of $\mathbf{v}_i$ and $\mathbf{u}_i$ is at least $C \log(d/\epsilon)$. Taking a union bound we get that the event $E_{\frac{\epsilon}{d}}$ holds with probability at least $1 - \epsilon$. On the event $E_{\frac{\epsilon}{d}}$, we obtain the following series of inequalities:

$$\sup_{\|\mathbf{w}\|_\infty \leq 1} \inf_{\mathbf{s},\mathbf{T}} \sup_{\|\mathbf{x}\|_\infty \leq 1} |\mathbf{w}^T \mathbf{x} - (\mathbf{s}_2 \odot \mathbf{v})^T \sigma((\mathbf{S}_1 \odot \mathbf{M})\mathbf{x})|$$

$$\leq \sup_{\|\mathbf{w}\|_\infty \leq 1} \inf_{\mathbf{s} \in \{0,1\}^{dn}} \sup_{\|\mathbf{x}\|_\infty \leq 1} |\mathbf{w}^T \mathbf{x} - (\mathbf{s}_2 \odot \mathbf{v})^T \sigma(\mathbf{M}'\mathbf{x})|$$

(Pruning $\mathbf{M}$ according to Step 1 (Pre-processing $\mathbf{M}$).)

$$= \sup_{\|\mathbf{w}\|_\infty \leq 1} \inf_{\mathbf{s}_1,\dots,\mathbf{s}_d \in \{0,1\}^n} \sup_{\|\mathbf{x}\|_\infty \leq 1} \left| \sum_{i=1}^{d} w_i x_i - \sum_{i=1}^{d} (\mathbf{s}_i \odot \mathbf{v}_i)^T \sigma(\mathbf{u}_i x_i) \right| \quad \text{(Using Eq. (10))}$$

$$\leq \sup_{\|\mathbf{w}\|_\infty \leq 1} \inf_{\mathbf{s}_1,\dots,\mathbf{s}_d \in \{0,1\}^n} \sup_{\|\mathbf{x}\|_\infty \leq 1} \sum_{i=1}^{d} |w_i x_i - (\mathbf{s}_i \odot \mathbf{v}_i)^T \sigma(\mathbf{u}_i x_i)|$$

$$= \sum_{i=1}^{d} \sup_{|w_i|\leq 1} \inf_{\mathbf{s}_i \in \{0,1\}^n} \sup_{|x_i|\leq 1} |w_i x_i - (\mathbf{s}_i \odot \mathbf{v}_i)^T \sigma(\mathbf{u}_i x_i)|$$

$$\leq \sum_{i=1}^{i} d\frac{\epsilon}{d} \qquad \text{(By definition of the event } E_{\frac{\epsilon}{d}})$$

$$\leq \epsilon.$$

$\square$

## A.2   Approximating a single layer

In this subsection, we approximate a layer from the target network by pruning 2 layers of a randomly initialized network. The overview of the construction is given in Figure 4.

Figure 4: Approximating a layer $\sigma(\mathbf{W}\mathbf{x})$: A diagram showing our construction to approximate a layer. Let $\mathbf{w}_1, \mathbf{w}_2, \ldots, \mathbf{w}_d$ be the $d$ rows of $\mathbf{W}$, i.e., the weights of $d$ neurons. Our construction has an additional hidden layer, which contains $d$ blocks (highlighted in blue), where each unit contains $k = O(\log(\frac{d}{\epsilon}))$ neurons. We first pre-process the weights by pruning the first layer so that it has a block structure as shown. For ease of visualization, we only show two connections per block, i.e., each neuron in the $i^{\text{th}}$ block is connected to $x_i$ and (before pruning) all the output neurons.

**Lemma 3.** *(Approximating a layer) Consider a randomly initialized two layer neural network* $g(\mathbf{x}) = \mathbf{N}\sigma(\mathbf{M}\mathbf{x})$ *with* $\mathbf{x} \in \mathbb{R}^{d_1}$ *such that* $\mathbf{N}$ *has dimension* $\left(d_2 \times Cd_1 \log \frac{d_1 d_2}{\epsilon}\right)$ *and* $\mathbf{M}$ *has dimension* $\left(Cd_1 \log \frac{d_1 d_2}{\epsilon} \times d_1\right)$, *where each weight is initialized independently from the distribution* $U[-1, 1]$.

*Let* $\widehat{g}(x) = (\mathbf{S} \odot \mathbf{N})^T \sigma((\mathbf{T} \odot \mathbf{M})\mathbf{x})$ *be the pruned network for a choice of pruning matrices* $\mathbf{S}$ *and* $\mathbf{T}$. *If* $f_{\mathbf{W}}(\mathbf{x}) = \mathbf{W}\mathbf{x}$ *is the linear (single layered) network, where* $\mathbf{W}$ *has dimensions* $d_2 \times d_1$, *then with probability at least* $1 - \epsilon$,

$$\sup_{\mathbf{W}:\|\mathbf{W}\|\leq 1, \mathbf{W}\in\mathbb{R}^{d_2 \times d_1}} \exists \mathbf{S}, \mathbf{T} : \sup_{\mathbf{x}:\|\mathbf{x}\|_\infty \leq 1} \|f_{\mathbf{W}}(\mathbf{x}) - \widehat{g}(\mathbf{x})\| < \epsilon.$$

*Proof.* Our proof strategy is similar to the proof in Lemma 2.

**Step 1: Pre-processing** $\mathbf{M}$  Similar to Lemma 2, we begin by pruning $\mathbf{M}$ to get a block diagonal matrix $\mathbf{M}'$.

$$\mathbf{M}' = \begin{bmatrix} \mathbf{u}_1 & 0 & \ldots & 0 \\ 0 & \mathbf{u}_2 & \ldots & 0 \\ \vdots & \vdots & \ldots & 0 \\ 0 & 0 & \ldots & \mathbf{u}_{d_1} \end{bmatrix}, \qquad \text{where } \mathbf{u}_i \in \mathbb{R}^{C\log\left(\frac{d_1 d_2}{\epsilon}\right)}$$

Thus, $\mathbf{T}$ is such that $\mathbf{M}' = \mathbf{T} \odot \mathbf{M}$. We also decompose $\mathbf{N}$ and $\mathbf{S}$ as following

$$\mathbf{S} = \begin{bmatrix} \mathbf{s}_{1,1}^T & \cdots & \mathbf{s}_{1,d_1}^T \\ \mathbf{s}_{2,1}^T & \cdots & \mathbf{s}_{2,d_1}^T \\ \vdots & \cdots & \vdots \\ \mathbf{s}_{d_2,1}^T & \cdots & \mathbf{s}_{d_2,d_1}^T \end{bmatrix}, \qquad \mathbf{N} = \begin{bmatrix} \mathbf{v}_{1,1}^T & \cdots & \mathbf{v}_{1,d_1}^T \\ \mathbf{v}_{2,1}^T & \cdots & \mathbf{v}_{2,d_1}^T \\ \vdots & \cdots & \vdots \\ \mathbf{v}_{d_2,1}^T & \cdots & \mathbf{v}_{d_2,d_1}^T \end{bmatrix}, \qquad \text{where } \mathbf{v}_{i,j}, \mathbf{u}_i \in \mathbb{R}^{C\log\left(\frac{d_1 d_2}{\epsilon}\right)}$$

Using this notation, we get the following relation:

$$(\mathbf{S} \odot \mathbf{N})\sigma(\mathbf{M}'\mathbf{x}) = \begin{bmatrix} \sum_{j=1}^{d_1}(\mathbf{s}_{1,j} \odot \mathbf{v}_{1,j})^T\sigma(\mathbf{u}_j x_j) \\ \vdots \\ \sum_{j=1}^{d_1}(\mathbf{s}_{d_2,j} \odot \mathbf{v}_{d_2,j})^T\sigma(\mathbf{u}_j x_j) \end{bmatrix} \tag{11}$$

**Step 2: Pruning N** Note that $\mathbf{v}_{i,j}$ and $\mathbf{u}_i$ contain i.i.d. random variables from Uniform distribution. Let $n = C\log(d_1 d_2/\epsilon)$ and define $E_{i,j,\epsilon}$ be the following event from the Lemma 1:

$$E_{i,j,\epsilon} := \left\{ \sup_{w\in[-1,1]} \inf_{\mathbf{s}_{i,j}\in\{0,1\}^n} \sup_{x:|x|\leq 1} |wx - (\mathbf{v}_{i,j} \odot \mathbf{s}_{i,j})^T\sigma(\mathbf{u}_i x)| \leq \epsilon \right\}$$

Define $E_\epsilon := \bigcap_{1\leq i\leq d_2} \bigcap_{1\leq j\leq d_1} E_{i,j,\epsilon}$ to be the intersection of all individual events. Lemma 1 states that each event $E_{i,j,\frac{\epsilon}{d_1 d_2}}$ holds with probability $1 - \frac{\epsilon}{d_1 d_2}$ because $\mathbf{u}_i$ and $\mathbf{v}_{i,j}$ have dimensions at least $C\log(\frac{d_1 d_2}{\epsilon})$. By a union bound, the event $E_{\frac{\epsilon}{d_1 d_2}}$ holds with probability $1 - \epsilon$. On the event $E_{\frac{\epsilon}{d_1 d_2}}$, we get the following inequalities:

$$\sup_{\mathbf{W}:\|\mathbf{W}\|\leq 1} \inf_{\mathbf{S},\mathbf{T}} \sup_{\|\mathbf{x}\|_\infty\leq 1} \|\mathbf{W}\mathbf{x} - (\mathbf{S}\odot\mathbf{N})^T\sigma((\mathbf{T}\odot\mathbf{M})\mathbf{x})\|$$

$$\leq \sup_{\mathbf{W}:\|\mathbf{W}\|\leq 1} \inf_{\mathbf{S}} \sup_{\|\mathbf{x}\|_\infty\leq 1} \|\mathbf{W}\mathbf{x} - (\mathbf{S}_2\odot\mathbf{N})^T\sigma(\mathbf{M}'\mathbf{x})\|$$

(Pruning $\mathbf{M}$ according to Step 1 (Pre-processing $\mathbf{M}$))

$$\leq \sup_{\mathbf{W}:\|\mathbf{W}\|\leq 1} \inf_{\mathbf{s}_{i,j}\in\{0,1\}^n} \sup_{\|\mathbf{x}\|_\infty\leq 1} \sum_{i=1}^{d_2}\left|\sum_{j=1}^{d_1}w_{i,j}x_j - \sum_{j=1}^{d_1}(\mathbf{s}_{i,j}\odot\mathbf{v_{i,j}})^T\sigma(\mathbf{u_j}x_j)\right|$$

(Using Eq. (11))

$$\leq \sup_{w_{i,j}:|w_{i,j}|\leq 1} \inf_{\mathbf{s}_{i,j}\in\{0,1\}^n} \sup_{x_j:|x_j|\leq 1} \sum_{i=1}^{d_2}\sum_{j=1}^{d_1}\left|w_{i,j}x_j - (\mathbf{s}_{i,j}\odot\mathbf{v_{i,j}})^T\sigma(\mathbf{u_j}x_j)\right|$$

$$\leq \sup_{w_{i,j}:|w_{i,j}|\leq 1} \inf_{\mathbf{s}_{i,j}\in\{0,1\}^n} \sum_{i=1}^{d_2}\sum_{j=1}^{d_1} \sup_{x_j:|x_j|\leq 1}\left|w_{i,j}x_j - (\mathbf{s}_{i,j}\odot\mathbf{v_{i,j}})^T\sigma(\mathbf{u_j}x_j)\right|$$

$$= \sum_{i=1}^{d_2}\sum_{j=1}^{d_1} \sup_{w_{i,j}:|w_{i,j}|\leq 1} \inf_{\mathbf{s}_{i,j}\in\{0,1\}^n} \sup_{x_j:|x_j|\leq 1}\left|w_{i,j}x_j - (\mathbf{s}_{i,j}\odot\mathbf{v_{i,j}})^T\sigma(\mathbf{u_j}x_j)\right|$$

$$\leq d_1 d_2 \frac{\epsilon}{d_1 d_2} \leq \epsilon. \qquad \text{(By definition of the event } E_{\frac{\epsilon}{d_1 d_2}})$$

$\square$

## A.3 Proof of Theorem 1

We now state the proof of Theorem 1 with the help of the lemmas in the previous subsection.

*Proof.* (Proof of Theorem 1) Let $\mathbf{x}_i$ be the input to the $i$-th layer of $f_{(\mathbf{W}_l,\ldots,\mathbf{W}_1)}(\mathbf{x})$. Thus,

1. $\mathbf{x}_1 = \mathbf{x}$,

2. for $1 \leq i \leq l-1$, $\mathbf{x}_{i+1} = \sigma(\mathbf{W}_i\mathbf{x}_i)$.

Thus $f_{(\mathbf{W}_l,\ldots,\mathbf{W}_1)}(\mathbf{x}) = \mathbf{W}_l\mathbf{x}_l$.

For $i^{th}$ layer weights $\mathbf{W}_i$, let $\mathbf{S}_{2i}$ and $\mathbf{S}_{2i-1}$ be the binary matrices that achieve the guarantee in Lemma 3. Lemma 3 states that with probability $1 - \frac{\epsilon}{2l}$ the following event holds:

$$\sup_{\mathbf{W}_i\in\mathbb{R}^{d_{i+1}\times d_i}:\|\mathbf{W}_i\|\leq 1} \exists \mathbf{S}_{2i},\mathbf{S}_{2i-1}: \sup_{\mathbf{x}:\|\mathbf{x}\|\leq 1} \|\mathbf{W}_i\mathbf{x} - (\mathbf{M}_{2i}\odot\mathbf{S}_{2i})\sigma((\mathbf{S}_{2i}\odot\mathbf{M}_{2i-1})\mathbf{x})\| < \epsilon/2l.$$

$$\tag{12}$$

As ReLU is 1-Lipschitz, the above event implies the following:

$$\sup_{\mathbf{W}_i \in \mathbb{R}^{d_{i+1} \times d_i} : \|\mathbf{W}_i\| \leq 1} \exists \mathbf{S}_{2i}, \mathbf{S}_{2i-1} : \sup_{\mathbf{x} : \|\mathbf{x}\| \leq 1} \|\sigma(\mathbf{W}_i \mathbf{x}) - \sigma((\mathbf{M}_{2i} \odot \mathbf{S}_{2i}) \sigma((\mathbf{S}_{2i} \odot \mathbf{M}_{2i-1}) \mathbf{x}))\| < \epsilon/2l.$$

(13)

Taking a union bound, we get that with probability $1 - \epsilon$, the above inequalities (12) and (13) hold for every layer simultaneously. For the remainder of the proof, we will assume that this event holds. For the any fixed function $f$, let $g_f = g_{(\mathbf{W}_l, \ldots, \mathbf{W}_1)}$ be the pruned network constructed layer-wise, by pruning with binary matrices satisfying Eq. (12) and Eq. (13), and let these pruned matrices be $\mathbf{M}'_i$. Let $\mathbf{x}'_i$ be the input to the $2i - 1$-th layer of $g_f$. We note that $\mathbf{x}'_i$ satisfies the following recurrent relations:

1. $\mathbf{x}'_1 = \mathbf{x}$,

2. for $1 \leq i \leq l - 1$, $\mathbf{x}'_{i+1} = \sigma(\mathbf{M}'_{2i} \sigma(\mathbf{M}'_{2i-1} \mathbf{x}'_i))$.

Because the input $\mathbf{x}$ has $\|\mathbf{x}\| \leq 1$, Equation (13) also states that $\|\mathbf{x}'_i\| \leq \left(1 + \frac{\epsilon}{2l}\right)^{i-1}$. To see this, note that we use Equation (13) to get for $1 \leq i \leq l - 1$ as

$$\|\sigma(\mathbf{W}_i \mathbf{x}'_i) - \mathbf{x}'_{i+1}\| \leq \|\mathbf{x}'_i\|(\epsilon/2l)$$
$$\implies \|\mathbf{x}'_{i+1}\| \leq \|\mathbf{x}'_i\|(\epsilon/2l) + \|\sigma(\mathbf{W}_i \mathbf{x}'_i)\| \leq \|\mathbf{x}'_i\|(\epsilon/2l) + \|\mathbf{W}_i \mathbf{x}'_i\| \leq \|\mathbf{x}'_i\|(\epsilon/2l) + \|\mathbf{x}'_i\|.$$

Applying this inequality recursively, we get the claim that for $1 \leq i \leq l - 1$, $\|\mathbf{x}'_i\| \leq \left(1 + \frac{\epsilon}{2l}\right)^{i-1}$. Using this, we can bound the error between $\mathbf{x}_i$ and $\mathbf{x}'_i$. For $1 \leq i \leq l - 1$,

$$\begin{aligned}
\|\mathbf{x}_{i+1} - \mathbf{x}'_{i+1}\| &= \|\sigma(\mathbf{W}_i \mathbf{x}_i) - \sigma(\mathbf{M}'_{2i} \sigma(\mathbf{M}'_{2i-1} \mathbf{x}'_i))\| \\
&\leq \|\sigma(\mathbf{W}_i \mathbf{x}_i) - \sigma(\mathbf{W}_i \mathbf{x}'_i)\| + \|\sigma(\mathbf{W}_i \mathbf{x}'_i) - \sigma(\mathbf{M}'_{2i} \sigma(\mathbf{M}'_{2i-1} \mathbf{x}'_i))\| \\
&\leq \|\mathbf{x}_i - \mathbf{x}'_i\| + \|\mathbf{W}_i \mathbf{x}'_i - \mathbf{M}'_{2i} \sigma(\mathbf{M}'_{2i-1} \mathbf{x}'_i)\| \\
&< \|\mathbf{x}_i - \mathbf{x}'_i\| + \left(1 + \frac{\epsilon}{2l}\right)^{i-1} \frac{\epsilon}{2l},
\end{aligned}$$

where we use Equation (12). Unrolling this we get

$$\|\mathbf{x}_l - \mathbf{x}'_l\| \leq \sum_{i=1}^{l-1} \left(1 + \frac{\epsilon}{2l}\right)^{i-1} \frac{\epsilon}{2l}.$$

Finally using the inequality above, we get that with probability at least $1 - \epsilon$,

$$\begin{aligned}
\|f_{(\mathbf{W}_l, \ldots, \mathbf{W}_1)}(\mathbf{x}) - g_{(\mathbf{W}_l, \ldots, \mathbf{W}_1)}(\mathbf{x})\| &= \|\mathbf{W}_l \mathbf{x}_l - \mathbf{M}'_{2l} \sigma(\mathbf{M}'_{2l-1} \mathbf{x}'_l)\| \\
&\leq \|\mathbf{W}_l \mathbf{x}_l - \mathbf{W}_l \mathbf{x}'_l\| + \|\mathbf{W}_l \mathbf{x}'_l - \mathbf{M}'_{2l} \sigma(\mathbf{M}'_{2l-1} \mathbf{x}'_l)\| \\
&\leq \|\mathbf{x}_l - \mathbf{x}'_l\| + \|\mathbf{W}_l \mathbf{x}'_l - \mathbf{M}'_{2l} \sigma(\mathbf{M}'_{2l-1} \mathbf{x}'_l)\| \\
&< \|\mathbf{x}_l - \mathbf{x}'_l\| + \left(1 + \frac{\epsilon}{2l}\right)^{l-1} \frac{\epsilon}{2l} \\
&\leq \left(\sum_{i=1}^{l-1} \left(1 + \frac{\epsilon}{2l}\right)^{i-1} \frac{\epsilon}{2l}\right) + \left(1 + \frac{\epsilon}{2l}\right)^{l-1} \frac{\epsilon}{2l} \\
&\leq \sum_{i=1}^{l} \left(1 + \frac{\epsilon}{2l}\right)^{i-1} \frac{\epsilon}{2l} \\
&= \left(1 + \frac{\epsilon}{2l}\right)^{l} - 1 \\
&< e^{\epsilon/2} - 1 \\
&< \epsilon. \qquad \text{(Since } \epsilon < 1.\text{)}
\end{aligned}$$

Replacing $\epsilon$ in this proof with $\min\{\epsilon, \delta\}$ gives us the statement of the theorem. $\qquad \square$

# B Proof of Lower Bound

*Proof.* (Proof xof Theorem 2) Firstly, note that $h_{\mathbf{W}}(\mathbf{x}) = \mathbf{W}\mathbf{x}$. Another fact we use in this proof is that matrices $\mathbf{W}$ of dimension $d \times d$ can be considered as points in the space $\mathbb{R}^{d \times d} \equiv \mathbb{R}^{d^2}$. The metric that we would be using on this space would be the operator norm of matrices $\| \cdot \|$. Note that $\mathcal{G}$ is a random set of functions, but we abuse the notation by using $|\mathcal{G}|$ denote the *maximum* number of sub-networks that can be formed, starting from any initialization with the given architecture.

**Step 1: Packing argument.** Consider the normed space of $d \times d$ matrices, $\mathcal{W} = \{\mathbf{W} \in \mathbb{R}^{d \times d} : \|\mathbf{W}\| \leq 1\}$, with the operator norm $\| \cdot \|$. Let $\mathcal{P}$ be a $2\epsilon$-separated set of $(\mathcal{W}, \| \cdot \|)$, i.e. $\mathcal{P} \subset \mathcal{W}$ and $\|\mathbf{M} - \mathbf{M}'\| > 2\epsilon$ for all distinct $\mathbf{M}, \mathbf{M}' \in \mathcal{P}$.

Note that any function $g'$ can only approximate at most one member of $\mathcal{P}$. To see this, let us assume on the contrary that a $g'$ can approximate two distinct members $\mathbf{W}_1$ and $\mathbf{W}_2$ of $\mathcal{P}$. Then a triangle inequality states that

$$\|\mathbf{W}_1 - \mathbf{W}_2\| = \sup_{\mathbf{x}:\|\mathbf{x}\|\leq 1} \|\mathbf{W}_1\mathbf{x} - \mathbf{W}_2\mathbf{x}\| \leq \sup_{\mathbf{x}:\|\mathbf{x}\|\leq 1} \|g'(\mathbf{x}) - \mathbf{W}_1\mathbf{x}\| + \sup_{\mathbf{x}:\|\mathbf{x}\|\leq 1} \|g'(\mathbf{x}) - \mathbf{W}_2\mathbf{x}\| \leq 2\epsilon,$$

which is a contradiction to the definition of a $2\epsilon$-separated set. Hence, $g'$ can approximate at most only one member of $\mathcal{P}$.

**Step 2: Relation between $|\mathcal{G}|$ and $|\mathcal{P}|$.** The goal of this step is to show that, under the theorem assumptions, $|\mathcal{P}| < 2|\mathcal{G}|$. If $|\mathcal{P}| > 2|\mathcal{G}|$, then we show that one of the matrices in $\mathcal{P}$ is the difficult matrix $W$ that we're looking for.

Let us assume that $|\mathcal{P}| > 2|\mathcal{G}|$. Recall that the previous step states that, for any realization of $g$, the corresponding $\mathcal{G}$ can only approximate at most $|\mathcal{G}|$ matrices in $\mathcal{P}$. Therefore, for a fixed realization of $\mathcal{G}$, we get that

$$\frac{\sum_{\mathbf{W} \in \mathcal{P}} \mathbb{I}\left(\exists g' \in \mathcal{G} : \sup_{\mathbf{x}:\|\mathbf{x}\|\leq 1} \|g'(\mathbf{x}) - \mathbf{W}\mathbf{x}\| \leq \epsilon\right)}{|\mathcal{P}|} \leq \frac{|\mathcal{G}|}{|\mathcal{P}|} < \frac{1}{2}.$$

Taking the expectation over the distribution of $g$, we get that

$$\frac{\sum_{\mathbf{W} \in \mathcal{P}} \mathbb{P}\left(\exists g' \in \mathcal{G} : \sup_{\mathbf{x}:\|\mathbf{x}\|\leq 1} \|g'(\mathbf{x}) - W\mathbf{x}\| \leq \epsilon\right)}{|\mathcal{P}|} < \frac{1}{2}.$$

As the minimum is less than the average, there exists a $\mathbf{W} \in \mathcal{P}$ such that $\mathbb{P}\left(\exists g' \in \mathcal{G} : \sup_{\mathbf{x}:\|\mathbf{x}\|\leq 1} \|g'(\mathbf{x}) - W\mathbf{x}\| \leq \epsilon\right) < \frac{1}{2}$, which is a contradiction to Eq. (9). Therefore, $2|\mathcal{G}| > |\mathcal{P}|$.

**Step 3: Lower bound on $|\mathcal{P}|$.** We will now choose $\mathcal{P}$ with the maximum cardinality of all $2\epsilon$-separated sets, i.e., that achieves the packing number. As packing number is lower bounded by the covering number, we will try to find a lower bound on the size of an $2\epsilon$-net of $\mathcal{W}$ [38, Lemma 4.2.8]. Now, any $2\epsilon$-cover has has to have at least $\frac{\text{Vol}(\{\mathbf{W}:\|\mathbf{W}\|\leq 1\})}{\text{Vol}(\{\mathbf{W}:\|\mathbf{W}\|\leq 2\epsilon\})}$ elements, where the volume is the Lebesgue measure in $\mathbb{R}^{d \times d} = \mathbb{R}^{d^2}$. We also have that $\text{Vol}(\{\mathbf{W} : \|\mathbf{W}\| \leq c\}) > 0$ because $\{\mathbf{W} : \|\mathbf{W}\| \leq c\}$ contains $\{\mathbf{W} : \|\mathbf{W}\|_{\text{Frobenius}} \leq c\}$. Thus, we get that $\frac{\text{Vol}(\{\mathbf{W}:\|\mathbf{W}\|\leq 1\})}{\text{Vol}(\{\mathbf{W}:\|\mathbf{W}\|\leq 2\epsilon\})} = (2\epsilon)^{-d^2}$. Putting everything together, we get that

$$2|\mathcal{G}| > |\mathcal{P}| > |\mathcal{N}(\mathcal{W}, \| \cdot \|, 2\epsilon)| \geq \left(\frac{1}{2\epsilon}\right)^{-d^2}.$$

**Case $l = 2$** Let the dimension of $\mathbf{M}_2$ be $d \times s$ and the dimension of $\mathbf{M}_1$ be $s \times d$. We need a lower bound on $s$. Now, the number of matrices that can be created by pruning $\mathbf{M}_2$ are $2^{sd}$ and similarly the number of matrices that can be created by pruning $\mathbf{M}_1$ are $2^{sd}$. Thus, the total number of ReLUs that can be formed by pruning $\mathbf{M}_2$ and $\mathbf{M}_1$ is at most $2^{2sd}$. Thus, $|\mathcal{G}| \leq 2^{2sd}$. Therefore, we get that

$$2^{2sd+1} > \left(\frac{1}{2\epsilon}\right)^{-d^2}.$$

This shows that $s = \Omega\left(d \log\left(\frac{1}{2\epsilon}\right)\right)$ is needed to approximate every function in $\mathcal{F}$ by pruning $g$ with probability $1/2$.

**Case $l > 2$**  Let the total number of parameters be $m$. Therefore, we get that $|\mathcal{G}| \leq 2^m$. Following the same arguments as before, we get that $m = \Omega\left(d^2 \log\left(\frac{1}{2\epsilon}\right)\right)$. $\qquad\square$

## C   Subset sum results

### C.1   Product of uniform distributions contains a uniform distribution

**Lemma 4.** *Let $X \sim U[0, 1]$ (or $X \sim U[-1, 0]$) and $Y \in U[-1, 1]$ be independent random variables. Then the PDF of the random variable $XY$ is*

$$
f_{XY}(z) = \begin{cases} \frac{1}{2} \log \frac{1}{|z|} & |z| \leq 1 \\ 0 & \textit{otherwise} \end{cases}
$$

*Proof.* It is easy to see why $f_{XY}(z) = 0$ for $z > 1$. We prove for $X \sim U[0, 1]$. The proof for $X \sim U[-1, 0]$ is similar.

Let us first try to find the CDF of $XY$.

Let $0 \leq z \leq 1$ be a real number. Note that $XY \leq 1$. Now, if $XY \leq z$, and if $Y \geq z$, then $X \leq z/Y$. However, if $Y < z$, then $X$ can be anything in its support $[0, 1]$. Thus,

$$
\begin{aligned}
F_{XY}(z) &= \mathbb{P}(XY \leq z) \\
&= \int_0^z \frac{1}{2} \int_0^1 1 \mathrm{d}x\mathrm{d}y + \int_z^1 \frac{1}{2} \int_0^{z/y} 1 \mathrm{d}x\mathrm{d}y \\
&= \frac{z}{2} + \frac{1}{2} \int_z^1 \frac{z}{y}\mathrm{d}y \\
&= \frac{z}{2} - \frac{z \log z}{2}.
\end{aligned}
$$

Differentiating this, the pdf for $0 \leq z \leq 1$ is

$$
f_{XY}(z) = \frac{1}{2} \log \frac{1}{z}.
$$

Now, because $XY$ is symmetric around 0, we get that for $|z| \leq 1$

$$
f_{XY}(z) = \frac{1}{2} \log \frac{1}{|z|}.
$$

$\qquad\square$

**Corollary 1.** *Let $X \sim U[0, 1]$ (or $X \sim U[-1, 0]$) and $Y \in U[-1, 1]$ be independent random variables. Let $P$ be the distribution of $XY$. Let $\delta_0$ be the Dirac-delta function. Define a distribution $D = \frac{1}{2}\delta_0 + \frac{1}{2}P$.*

*Then, there exists a distribution $Q$ such that*

$$
P = \left(\frac{1}{2} \log 2\right) U\left[-\frac{1}{2}, \frac{1}{2}\right] + \left(1 - \frac{1}{2} \log 2\right) Q
$$

*Proof.* The corollary follows from the observation that Lemma 4 shows that pdf of $P$ is lower bounded by $(\log 2)U\left[-\frac{1}{2}, \frac{1}{2}\right]$ on $\left[-\frac{1}{2}, \frac{1}{2}\right]$. $\qquad\square$

### C.2   Subset sum problem with product of uniform distributions

**Corollary 2** ( [31]). *Let $X_1, \ldots, X_n$ be i.i.d. from the distribution in the hypothesis of Corollary 1, where $n \geq C \log \frac{2}{\epsilon}$ (for some universal constant $C$). Then, with probability at least $1 - \epsilon$, we have*

$$
\forall z \in [-1, 1], \qquad \exists S \subset [n] \text{ such that } \left| z - \sum_{i \in S} X_i \right| \leq \epsilon.
$$

*Proof.* This is a direct application of Markov's inequality on Corollary 3.3 from [31] applied to the distribution in the hypothesis of Corollary 1. □