[Reviews · NeurIPS 2020]

Review 1

Summary and Contributions: This paper considers the strong lottery ticket hypothesis -- a conjecture that a randomly initialized neural network can be pruned (i.e. only have weights removed) in order to achieve good performance against some target function. The authors show that when the target function is a fully-connected neural network, such a pruning will exist with high probability whenever the randomly initialized network has twice as many layers and has a width that is a log(d*l/epsilon) factor larger than the target network. Here, d is the width of the target network, l is its depth, and epsilon is the desired accuracy. This is an improvement over the best/only known result (Malach et al. 2020) on this problem that showed that this can be achieved with a width that is a poly(d, l, 1/epsilon) factor larger than the target. The improvement is achieved by essentially reusing the proof of Malach et al., but fixing a key step where polynomial factors were lost by appealing to known results on random subset sum problems. The authors also provide a lower bound showing that approximating a two layer network with a constant layer pruned network will require a log(1/epsilon) factor blowup in width.

Strengths: 1. Tightest known bounds on the strong lottery ticket hypothesis. 2. Accompanying lower bounds showing that a log(1/epsilon) blowup in the number of parameters is required when the target network is a two layer network. 3. The improvement over Malach et al. is simple and understandable.

Weaknesses: It is hard to find a glaring weakness here. This paper identifies the issue in the only known result on the lottery ticket hypothesis and fixes it by appealing to a previous result on the subset sum problem.

Correctness: The paper appears to be correct.

Clarity: The paper is clear and well-written.

Relation to Prior Work: This paper adequately explains the previous state of the art on the problem and the improvements provided in the paper.

Reproducibility: Yes

Additional Feedback: After reading the author response and the other reviews, I still feel that my score is appropriate.


Review 2

Summary and Contributions: This paper studies the problem of lottery ticket hypothesis. They theoretically show that any fully connected target network can be approximated by pruning an random one within only only log factor of over-parameterization. The key idea is to connect it to the SUBSETSUM problem. Experiments are also provided to justify their results.

Strengths: The paper improves previous result from a polynomial factor of over-parameterization to a log factor of over-parameterization. They also provide lower bound for the required over-parameterization, which indicates their upper bound is optimal up to log factor. These results improve the understanding of lottery ticket hypothesis, which is currently a very interesting topic.

Weaknesses: The construction of subnetwork given by this paper relies on the solving SUBSETSUM, which is a NP-hard problem in general. So, it might be time-consuming in practice. It would be interesting to find more efficient way to find such subnetwork, as mentioned by authors.

Correctness: The claims and experiments seem to be correct, from my point of view.

Clarity: The paper overall is well-written and easy to understand.

Relation to Prior Work: The authors discussed related works, and explained several differences between prior works and current paper.

Reproducibility: Yes

Additional Feedback: In the experiments, when using the network structure in Figure 1 and pruning it by solving SUBSETSUM directly or using other method (e.g. edge-popup in Experiment 2), I was wondering whether the subnetwork we get would be similar, or the number of parameters left would be different. ============================================================= After rebuttal: Thanks for the response, I would like to keep my score.


Review 3

Summary and Contributions: This paper studies the strong lottery ticket hypothesis which states that one can approximate any target neural network by only pruning the weights of a sufficiently over-parameterized random network. This paper establishes a connection between pruning random networks and random instances of the SUBSETSUM problem and uses this connection to derive a new approach that exponentially improves the previous bounds in terms of the amount of over-parameterization. The paper complements with this positive result with a lower bound for two-layer neural networks and experiments.

Strengths: 1. Pruning is a very hot and important topic in DL. 2. This paper is very well-written. 3. The theoretical finding is very interesting. The use of random instances of the SUBSETSUM problem is novel in the literature. I believe this connection maybe useful in other compression problems in ML. 4. The exponential improvement over existing result is significant.

Weaknesses: The lower bound only applies to two-layer neural networks, though I think the lower bound is not the focus of this paper.

Correctness: Yes.

Clarity: Yes. I espeically appraciate the intuitive explaination of the proof.

Relation to Prior Work: Yes.

Reproducibility: Yes

Additional Feedback: Thanks for the response, and I'll keep my score. -------------------------------------------- Does the proof apply to other activation function? Like softplus? Is [1] a concurrent work? If so, I think author(s) should point it out more explicitly. Line 485: "xof" -> "of"


Review 4

Summary and Contributions: The paper tightens the bound on how over-parameterized a network has to be for a subset of its weights to approximate a target network. The work is a solid theoretical advance with respect to the lottery ticket hypothesis. The central idea is simple and clever, especially so because it leverages an obscure, 20+ year old theorem which provides an excellent improvement on the bound.

Strengths: Simple, elegant idea and extension of the pruning approach for bounding approximating network sizes for the lottery ticket hypothesis. My background is not that strong in this area, and my understanding of proof was only superficial, so others should evaluate the technical aspects to be sure.

Weaknesses: I wonder whether it's important to experiment with training the target network on other problems besides MNIST.

Correctness: As far as I understood them, yes.

Clarity: The paper is very clearly written. I understood everything up to Section 3.2, and I followed the key proof starting around line 189, but I didn't understand it deeply. My lack of comprehension is due to my lack of background in this area, and in general I think an expert will find it clear.

Relation to Prior Work: Yea, it builds directly on a recent result on bounding the network size with polynomial depth/layer size complexity, offers an explanation for why the previous version's complexity was as it is, and motivated their work by pointing to the Lueker paper and explaining how the subset sum strategy provides an advantage.

Reproducibility: Yes

Additional Feedback:

[Author Response · NeurIPS 2020]

We thank all the reviewers for their time and effort in providing feedback. We are encouraged by the universally positive scores (7 7 7 6) and that all the reviewers appreciated the paper for the following: (i) theoretical contributions (**R1**,**R2**,**R3**,**R5**), (ii) advancing our understanding of the LTH (**R2**,**R3**), (iii) novel connection to SubSetSum(**R3**,**R5**), (iv) clear exposition (**R1**,**R2**,**R3**,**R5**), and (v) relation with prior work (**R1**,**R2**,**R3**,**R5**). Moreover, **R3** thinks this connection with SubsetSum can be applicable in other areas.

For clarity, we would like to reiterate the goal and motivation of the paper. We provide theoretical justifications for the striking empirical observations in Ramanujan et al. [1]: Do good subnetworks *provably* exist with a small factor of overparameterization? Our main theoretical contribution in this paper is to *characterize* the required over-parameterization (up to logarithmic factors) for fully connected networks, offering **an exponential improvement** in the oveparameterization bounds by Malach et al.

We address the individual concerns below. We thank **R3** for pointing out the typo.

**SubsetSum is NP-Hard (R2):**   We note that the connection to the SubsetSum problem is made so we can establish an **existential** rather than an algorithmic result: in this work we prove the existence of a subnetwork that performs as well as any target network, but do not claim to offer a "good algorithm" to do so. In general, we do not have access to the target network, but only to the labeled training data. Thus, an important–yet orthogonal to our goal in this paper–question is whether there exist poly-time pruning algorithms with provable performance. As optimizing ReLU neural network is itself NP-Hard in general, we expect all algorithms to be inefficient in the worst case. However, similar to the success of backpropagation on standard tasks, the findings of [1] suggest that a pruning-based algorithm is potentially practical. Understanding the effectiveness of pruning given certain data/model assumptions is an important future direction that we are currently working on.

**Experimental comparison of target, subsetsum, and edge-popup pruned models (R2):**   We believe that a significant level of network isomorphism between the two pruned networks (obtained from SubsetSum and edge-popup) would be extremely unlikely, because the pruning algorithms are quite different. Because the networks differ in both weight distribution and structure, we cannot hope for a significant level network isomorphism (even upto the ordering of neurons in a layer), but we can still compare using some other metrics like sparsity (Figure 2 in our paper). For a two-layer fully-connected network, our SubsetSum approximation utilized $\sim 3.5$ million out of $\sim 8.3$ million coefficients in total. Thus, the approximated network achieved 97.17% test set accuracy with $\sim 44.69\%$ sparsity. On the other hand, one of our networks resulting from edge-popup achieved a 97.53% test set accuracy by retaining $\sim 0.5$ million of $\sim 2$ million parameters, thus giving $\sim 25\%$ sparsity, but on a much smaller network.

**Lower bound for deeper networks (R3):**   We would like to emphasize that our lower bound controls the number of parameters and thus is valid for any depth and architecture. In particular, for constant depth greater than 2, we directly obtain $d\sqrt{\log 1/\epsilon}$ lower bound on width. We agree that obtaining a tighter lower bound is an important question, and we believe it should be possible with regards to the way that the depth and width appears in the logarithm. Answering this question requires understanding the role of depth in the representation power of neural networks. This is a notoriously difficult problem and, even after years of research, remains elusive [2].

**Other activation functions (R3):**   Although our paper (and prior work [3]) focuses on ReLU activations, the proof strategy works for a general class of Lipschitz activations with a slight modification in the architecture. Specifically, following the structure of Section 3.1, suppose that every alternate layer is linear. The proof then goes through *exactly* as before by using the Lipschitz property in Eq. (13) (Line 468) in Appendix. We will add a comment regarding this in the final version of the paper.

**Experiments on other datasets (R5):**   We agree that experiments with datasets beyond MNIST would be beneficial to further validate our findings. Currently, our "diamond" based architecture inspired by Lueker's random SubsetSum theorem is defined only for fully connected layers. Data sets like CIFAR10 and ImageNet require deeper convolutional networks that contain very few fully-connected layers. A good test set accuracy on MNIST is obtainable with both shallow fully connected networks and LeNet5, which has more fully connected layers than convolutional layers. Thus, although somewhat limited, we believe that MNIST is an appropriate choice for experiments in this case. However, deep convolutional networks such as ResNet and VGG generally contain up to three fully connected layers that our structure shown in Figure 1c could be applied to, so we think that investigating the performance of our structure on deep convolutional networks is valid and worth further investigation. We were not able to complete the design and execution of these experiments by the rebuttal deadline, but are eager to do so for the camera ready version.

We would again like to thank the reviewers for the positive reviews.

## References

[1] V. Ramanujan, M. Wortsman, A. Kembhavi, A. Farhadi, and M. Rastegari. What's Hidden in a Randomly Weighted Neural Network? *arXiv:1911.13299*.

[2] G. Vardi and O. Shamir. Neural Networks with Small Weights and Depth-Separation Barriers. *arXiv:2006.00625*.

[3] E. Malach, G. Yehudai, S. Shalev-Shwartz, and O. Shamir. Proving the Lottery Ticket Hypothesis: Pruning is All You Need. *arXiv:2002.00585*.


[Meta-Review · NeurIPS 2020]

The reviewers reached a consensus that this paper deserves acceptance to neurips.